# High-Performance Dual-Ion Battery Based on Silicon–Graphene Composite Anode and Expanded Graphite Cathode

**DOI:** 10.3390/molecules28114280

**Published:** 2023-05-23

**Authors:** Guoshun Liu, Xuhui Liu, Xingdong Ma, Xiaoqi Tang, Xiaobin Zhang, Jianxia Dong, Yunfei Ma, Xiaobei Zang, Ning Cao, Qingguo Shao

**Affiliations:** School of Materials Science and Engineering, China University of Petroleum (East China), Qingdao 266580, China; z22140037@s.upc.edu.cn (G.L.);

**Keywords:** dual-ion battery, anode materials, Si nanospheres, energy density

## Abstract

Dual-ion batteries (DIBs) are a new kind of energy storage device that store energy involving the intercalation of both anions and cations on the cathode and anode simultaneously. They feature high output voltage, low cost, and good safety. Graphite was usually used as the cathode electrode because it could accommodate the intercalation of anions (i.e., PF_6_^−^, BF_4_^−^, ClO_4_^−^) at high cut-off voltages (up to 5.2 V vs. Li^+^/Li). The alloying-type anode of Si can react with cations and boost an extreme theoretic storage capacity of 4200 mAh g^−1^. Therefore, it is an efficient method to improve the energy density of DIBs by combining graphite cathodes with high-capacity silicon anodes. However, the huge volume expansion and poor electrical conductivity of Si hinders its practical application. Up to now, there have been only a few reports about exploring Si as an anode in DIBs. Herein, we prepared a strongly coupled silicon and graphene composite (Si@G) anode through in-situ electrostatic self-assembly and a post-annealing reduction process and investigated it as an anode in full DIBs together with home-made expanded graphite (EG) as a fast kinetic cathode. Half-cell tests showed that the as-prepared Si@G anode could retain a maximum specific capacity of 1182.4 mAh g^−1^ after 100 cycles, whereas the bare Si anode only maintained 435.8 mAh g^−1^. Moreover, the full Si@G//EG DIBs achieved a high energy density of 367.84 Wh kg^−1^ at a power density of 855.43 W kg^−1^. The impressed electrochemical performances could be ascribed to the controlled volume expansion and improved conductivity as well as matched kinetics between the anode and cathode. Thus, this work offers a promising exploration for high energy DIBs.

## 1. Introduction

Despite the high energy and power density of lithium ion batteries, the limited and uneven distribution of lithium and rare metal resources have led to the search of new energy storage technologies with low cost, high safety, and reliability [1,2,3]. DIBs are energy storage devices that store energy involving the intercalation of both anions and cations on the cathode and anode simultaneously. They feature high output voltage, low cost, and good safety [4]. Graphite was commonly used as a cathode electrode because it could accommodate intercalation of anions (i.e., PF_6_^−^, BF_4_^−^, ClO_4_^−^) at a high cut-off voltage (up to 5.2 V vs. Li^+^/Li) [5,6]. However, the tested capacity of graphite cathodes usually ranged between 80–130 mAh g^−1^, and it is difficult to be further increased. As for the anode side, any materials that can reversibly store cations can be used as anodes for DIBs [7]. In this case, the traditional graphite anode could be exchanged to other electrode materials with larger capacities. Recently, it was found that the replacement of the low capacity of graphite anodes with other large capacity anodes could indeed increase the capacity of the full DIBs. For instance, Wei et al. [8] assembled DIBs employing the high lithium storage capacity of MoSe_2_/Nitrogen-Doped Carbon (1224 mA h g^−1^) as an anode and graphite as a cathode, and the DIBs delivered a reversible discharge capacity of 86 mA h g^–1^ at 2C after 150 cycles. Tang et al. [9] studied Al foil (theoretical lithium storage capacity is 2235 mAh g^−1^) as an anode, and, when coupled with graphite cathode, the DIBs showed a reversible capacity of ≈100 mAh g^−1^ and a capacity retention of 88% after 200 charge–discharge cycles. Wang et al. [10] designed a high capacity Ge/CNFs anode (2614 mA h g^−1^), and the assembled Ge/CNFs-Graphite DIBs showed a high discharge capacity of 281 mA h g^−1^ at a discharge current of 0.25 A g^−1^, which greatly surpassed those of most of the reported DIBs. Therefore, in order to take full advantage of the DIBs, further optimization of the anode materials is also important.

According to the cation storage mechanisms, the anode materials can be divided into three types: the intercalation type, conversion type, and alloying type. However, the intercalation-type anode (such as graphite, soft carbon, hard carbon, Na_2_Ti_3_O_7_, etc.) usually exhibits low capacity, and the conversion-type anode (such as MoS_2_, MoSe_2_, Co_3_O_4_, etc.) suffers from low reaction kinetics and shows unimpressed rate performance. The alloying-type anode (such as Al, P, Si, Sn, Ge, etc.) reacts with cations to boost an extreme specific capacity [11,12,13]. The silicon anode materials possess a high theoretic Lithium storage capacity of 4200 mAh g^−1^ [14,15,16]. Therefore, it is an efficient method to improve the energy density of DIBs by combining graphite cathode with high-capacity silicon anode. However, the huge volume expansion (>300%) of silicon throughout alloying and de-alloying must be controlled [17,18,19,20]. Moreover, the poor electrical conductivity of silicon is also an important factor that hinders its development [21,22,23]. Up to now, there are only a few reports about exploring Si as an anode in DIBs. Shao et al. [24] prepared a Si/C anode by adding a little amount of Si (7.6 wt%) into graphite and used it in full DIBs with enhanced energy densities. Wang et al. [25] investigated pre-lithiated Si as an anode with the intention of tailoring the voltage to match the cathode. Tang et al. [26] cleverly designed a flexible interface between Si and a conducting soft polymer substrate to modulate the alloying stress of the silicon anode in DIBs, achieving excellent flexible electrochemical properties. Despite the above efforts, to fully take advantage of the large capacity of an Si anode, it is still a challenge to further design a new Si-based anode material with a controlled volume expansion effect and improved conductivity, as well as matched kinetics with the cathode side.

Herein, we prepared a strongly coupled silicon and graphene composite (Si@G) anode through in-situ electrostatic self-assembly and a post-annealing reduction process and investigated it as an anode in full DIBs. Home-made expanded graphite (EG) with a larger inter-layer distance was employed as a cathode with fast intercalation kinetic to accommodate anions (Figure 1). Half-cell tests showed that the as-prepared Si@G anode could retain a maximum specific capacity of 1182.4 mAh g^−1^ after 100 cycles, whereas the bare Si anode only maintained a low capacity of 435.8 mAh g^−1^. Moreover, the full Si@G//EG DIBs achieved a high energy density of 367.84 Wh kg^−1^ at a power density of 855.43 W kg^−1^. The impressed electrochemical performances could be ascribed to the following aspects: (i) the composite structure greatly suppressed the stress/strain induced by volume change and alleviated the pulverization during charging and discharging; (ii) the graphene layer adhered on the surface of Si nanospheres could weaken volume expansion and prevent the inter-agglomeration of Si with high surface energy; (iii) the electron migration during the charge/discharge process was promoted by highly conductive graphene; (iv) the improved rate ability of the Si@G anode matched well with the fast kinetics of the EG cathode.

## 2. Results and Discussion

### 2.1. Structure and Morphology Analysis

Figure 2a is the XRD comparison of Si@G-1, Si@G-5 and Si@G-10. Evident diffraction peaks are observed at 28.46, 47.26, 56.20, 69.12, and 76.28°, corresponding to the (111), (220), (311), (400), and (331) crystallographic planes of silicon, respectively. The characteristic diffraction peaks of Si@G-5 and Si@G-10 correspond almost exactly to the standard cards, and almost no diffraction peaks of graphene appear, which is attributed to the comparatively low content of graphene in the composite. The weak peak of Si@G-1 at 25.58° corresponds to the (002) characteristic crystallographic plane of the graphene, implying that GO has been successfully reduced, and the product is free of byproducts such as SiC [27,28]. The spectrogram of the unreduced GO is shown in Figure 2b. After self-assembly, the graphene bonded well to silicon during the high-temperature reduction process, which mainly showed the characteristic peaks of Si. Figure 2c shows the Raman plot of composites. Si@G-1, Si@G-5, and Si@G-10 nanocomposites all have characteristic peaks at about 511 and 942.7 cm^−1^, corresponding to the characteristic peaks of nano-silicon. For Si@G-1, relatively weak graphene characteristic peaks at 1360 and 1580 cm^−1^ were observed. At the same time, no other by-products appeared. The whole nanocomposite exhibited high crystallinity.

Figure 3a shows the nitrogen adsorption and desorption curves of Si@G-1, Si@G-5, and Si@G-10, and it is evident that the curves all show typical type IV isotherms. The climbing of the H_3_ hysteresis loop at a relative pressure (P/Po) of 0.8 to 1.0 is evidence of the presence of mesoporous structures, which may originate from the voids between the adjacent graphene. Apparently, the specific surface area of Si@G-1 is higher than that of Si@G-5 and Si@G-10 due to the relatively more disordered arrangement of graphene. The specific surface areas of Si@G-5 and Si@G-10 are comparable, and the less graphene content cannot change the silicon arrangement. In addition, the pore diameter distribution curves (Figure 3b) show that the pore diameter is mainly around 5 nm, which is favorable for the migration of Li^+^ to Si@G inside the composites [29].

To characterize the morphology of the Si@G composites, SEM was performed. Figure 4a–c show the morphology of Si@G-1 sample. As can be observed, silicon nanoparticles (range of 50–100 nm) were anchored on graphene nanosheets, and the size of the graphene sheets was between 1 and 10 um. The silicon nanoparticles were completely encapsulated by the graphene nanosheets, and the agglomeration of silicon and the folding and stacking of graphene sheets created many voids that facilitated electrolyte penetration and Li^+^ transport [30,31]. At the same time, the silicon-loaded nanosheets were interconnected to provide additional electronic pathways. Figure 4d–f show the SEM images of the Si@G-5 composites, where more silicon nanoparticles mask the presence of graphene, but the graphene sheets can still be observed by higher magnification, as shown in Figure 4f. For the Si@G-10 sample, the extremely high content of silicon makes graphene not observable in the SEM images.

TEM was applied to further examine the structure of the Si@G-5 material. TEM images (Figure 5a,b) clearly demonstrate the presence of graphene sheets and silicon nanoparticles, with a few layers of large diameter graphene completely wrapping the silicon nanoparticles. It is evident that silicon nanoparticles are firmly anchored on graphene sheets due to the electrostatic binding effect. Figure 5c shows the HRTEM image of the silicon–graphene composite. The spacing of (111) is 0.31 nm, and a sheet of graphene tightly attached to the Si shells and amorphous carbon deposited on the surface of the composite can also be observed from the HRTEM image. From the dark-field diffraction image (Figure 5d), it is possible to derive each crystallographic plane of the Si nanoparticles, which confirms that the crystallinity of the Si nanoparticles is mostly preserved.

### 2.2. Electrochemical Studies

Structure and morphology analysis indicated that the Si@G-5 composite exhibited high crystallinity, proper specific surface area, and better integration. Thus, it was chosen to be tested as an electrode for the lithium half-cell, lithium-ion full cell, and dual-ion full battery. The half-cell was firstly assembled to evaluate the lithium storage capacity of the Si@G composite electrode using Si@G as the working electrode and lithium metal as the reference electrode. The CV curves of Si@G-5 are displayed in Figure 6a. Two cathodic peaks appeared in the first cycle. The broad peak at 0.69 V was attributed to the formation of the solid electrolyte interphase (SEI) and the contribution of graphene sheets [32,33]. Another cathodic peak out near 0.23 V was attributed to the transformation of the silicon structure from crystalline to amorphous [34]. In contrast, the two anodic peaks at 0.39 and 0.52 V corresponded to the de-lithiation process of amorphous Li_x_Si to Si.

Figure 6b shows the first three GCD curves of Si@G-5 anode at 0.1 A g^−1^. Due to the formation of SEI films, the ICE is 69.94%. In the second and third cycles, the curves overlap almost completely, and the specific discharge capacity is about 4100 mAh g^−1^. This is consistent with the CV results, indicating that the Si@G-5 has excellent reversibility. Figure 6c shows the curve of the 100 cycles of nano-silicon and Si@G-5 with the same current density. Apparently, the Si@G-5 material had a specific capacity of 1182.40 mAh g^−1^ after 100 cycles, with a capacity retention of 59.12%, much higher than the pure silicon sample of 435.8 mAh g^−1^ and 20.94%. Figure 6d shows the GCD curves under different cycle numbers. Except for the low CE of the first cycle, the subsequent CEs are higher than 95%, indicating that Si@G-5 has excellent stability and electrochemical reversibility. This is due to the strong binding force produced by electrostatic action, which makes the volume expansion of Si@G-5 in the cycle process is significantly inhibited [35].

The rate performance of pure silicon and Si@G-5 were tested, as shown in Figure 6e. At 0.1, 0.2, 0.5, 1.0, and 2.0 A g^−1^, the discharge capacities were 3348.8/3520.3, 2567.1/2874.4, 1999.5/2369.4, 1672.1/1973.4, and 1390/1600 mAh g^−1^, respectively. Evidently, Si@G-5 performed better than pure silicon samples at each current density. When the current density was restored to 0.1 A g^−1^ again, the Si@G-5 specific capacity was restored to 2801.6 mAh g^−1^, the capacity retention was 83.66%, and the pure silicon sample was only 60.09%. Figure 6f shows the charge–discharge curves of Si@G-5 under different current densities. As the current density increased, the specific capacity decreased. However, the voltage difference between charging and discharging platforms was still small, which indicated that Si@G-5 had small polarization, high reversibility, and electrode stability. These excellent properties could be attributed to the introduction of graphene to enhance the charge transfer ability, and the graphene sheets could effectively inhibit the internal stress caused by the silicon volume expansion.

In addition to the excellent rate performance, the Si@G-5 electrode also demonstrated excellent long-term cycling stability. As shown in Figure 7, after 1000 cycles at 1.0 A g^−1^, the reversible specific capacity remained at 325.1 mAh g^−1^, while the pure silicon sample failed after 1000 cycles due to volume expansion [36].

To verify the performance of the Si@G-5 in the DIB, the full cell of a lithium-based DIB was assembled, with Si@G-5 as the anode and (expanded graphite) EG as the cathode (Figure 8a). EG was selected because it possessed an enhanced interlaying distance, which could accommodate more PF6^−^ intercalation. The detailed structure and electrochemical PF6^−^ storage performances of the EG are shown in Appendix A. Figure 8b shows the CV curves of the EG//Si@G-5 DIB, which almost entirely overlap, indicating the excellent electrochemical reversibility. Three evident oxidation peaks appeared around 4.29, 4.60, and 4.83 V, which were the initial electrochemical processes occurring jointly at the cathode and anode, corresponding to different intercalation processes of PF6^−^ [37]. The first three cycles of the GCD curve of the EG//Si@G-5 DIB at 1 C are shown in Figure 8c, and the first charging specific capacity was up to 388.23 mAh g^−1^. However, the initial coulomb efficiency (ICE) was 24.31%, due to the formation of SEI films and side reactions [38]. Thereafter, the curves converge in terms of specific capacity and CE. The rate performance of the full cell is shown in Figure 8d. After 10 cycles at 2, 4, 6, 8, 10, 20, and 2 C in sequence, the discharge specific capacity recovered to 64.66 mAh g^−1^, with a capacity retention of 74.39%, and all CEs were around 95%. Figure 8e shows the GCD curves at different current densities. The cells have evident and similar charging and discharging plateaus, which correspond to the results of the CV curves. The capacity was 48.35 mAh g^−1^ after 100 cycles at 2 C, with a capacity retention of 57.18% (Figure 8f), and the capacity fade was also relatively slow. These were attributed to the synergistic effect of the cathode and anode, emphasizing the improved structure of the silicon-based electrode in the DIBs.

In addition, we also assembled a LiFePO_4_//Si@G-5 LIB full cell to compare the electrochemical performance of the LIB and DIB. The GCD curves and cycling curves of LiFePO_4_//Si@G-5 LIB are shown in Appendix A, respectively. The results show that its electrochemical performance was much less than EG//Si@G-5 DIB.

To further study the kinetic properties of the EG//Si@G-5 DIB, EIS was performed. Figure 9a shows the EIS plots of the EG//Si@G-5 DIB in the uncycled state, after 50 cycles, and after 200 cycles. The charge transfer resistance (R_ct_) was 34.03 Ω when uncycled (Figure 9b). After cycling, there was a different degree of increase in R_ct_ (Figure 9c). This may have been due to the creation of SEI films in the first cycle and was also responsible for the irreversible capacity and cell performance degradation in the first cycle [39]. In the low-frequency region, the curve flattened out, owing to the decreasing rate of diffusion. However, a high ion migration rate could be maintained after 200 cycles. The fitted curves were calculated as σ_0_ = 170.51, σ_50_ = 669.40, and σ_200_ = 1328.49, resulting in Li^+^ diffusion coefficients of 1.37 × 10^−21^, 8.90 × 10^−23^, and 2.26 × 10^−23^ cm^2^ s^−1^ for the three, respectively (please refer to Appendix A for the detailed calculation procedure).

To evaluate the value of the EG//Si@G-5 DIB for practical applications, we calculated the energy density and power density at different current densities (Appendix A). The EG//Si@G-5 DIB achieved an energy density of 367.84 Wh kg^−1^ at 2 C, while the LiFePO_4_//Si@G-5 LIB showed only a low energy density of 93.58 Wh kg^−1^. Moreover, the EG//Si@G-5 DIB could maintain a high energy density of 138.37 Wh kg^−1^ even at 20 C. These results demonstrated that the EG//Si@G-5 DIB held great potential for a high-performance energy storage device.

## 3. Materials and Methods

### 3.1. Chemicals and Reagents

For the synthesis of Si@G and EG materials, the following reagents were used: silicon powder (Si, cell grade), purchased from Shenzhen Huaqing Material Technology Co. (Shenzhen, China); natural graphite (C, purity ≥ 99%), purchased from Shenzhen Kejing Technology Co. (Shenzhen, China); Poly dimethyl diallyl ammonium chloride ((C_8_H_16_ClN)_n_, AR), sodium nitrate (NaNO_3_, AR), concentrated sulfuric acid (H_2_SO_4_, AR), potassium permanganate (KMnO_4_, AR), hydrogen peroxide (H_2_O_2_, AR), and anhydrous ethanol (C_2_H_6_O, AR), all the above chemical reagents were purchased from Sinopharm Chemical Reagent Co. (Shanghai, China). All chemical reagents used in this work were not further purified.

### 3.2. Material Preparation

#### 3.2.1. Preparation of Si@G Anode Material

First, 100 mg Si powder was weighed and fully stirred in 50 mL deionized water for 30 min, and then 2 mL 20 wt% PDDA solution (Poly dimethyl diallyl ammonium chloride) was added. The solution was stirred and ultrasonically dispersed for 1 h. Then, the above solution was centrifuged at 10,000 rpm for 5 min, and the excess PDDA was removed three times to obtain a positively charged Si precipitate. In total, 100 mg of graphite oxide (GO) was weighed, 100 mL of deionized water was added, mixed, stirred, and sonicated for 1 h to obtain the GO solution (1.0 mg mL^−1^).

The prepared aqueous Si-PDDA solution was slowly added to the GO solution and stirred for 12 h. The solution was filtered to obtain Si@GO filter cake, which was then dried under vacuum at 80 °C for 12 h. The dried filter cake was ground into powder and then calcined at 900 °C under 10% H_2_/Ar atmosphere for 2 h. Finally, the Si@G-1 sample (for comparison, the addition of 500 mg and 1000 mg of silicon powder were noted as Si@G-5 and Si@G-10) was obtained. As illustrated in Figure 1, the fabrication process is clearly shown.

#### 3.2.2. Preparation of EG Cathode Material

In total, 1 g of natural graphite and 0.5 g of NaNO_3_ were weighed and slowly added to 23 mL of concentrated H_2_SO_4_, while the temperature was controlled to below 5 °C. After stirring in a water bath for 30 min, the temperature was increased to 35 °C, and then 0.5 g of KMnO_4_ of different masses was added and kept stirred for 2 h. After that, 46 mL of deionized water was added, heated to 98 °C, and stirred continuously for 30 min. In total, 130 mL of deionized water and 10 mL of H_2_O_2_ were added.

The above solution was added to anhydrous ethanol and deionized water and continuously filtered at least four times to make the pH neutral. The filter cake was transferred to a vacuum oven and dried at 80 °C for 12 h. Subsequently, it was heated to 600 °C in N_2_ atmosphere and held for 5 h. Finally, EG was obtained.

### 3.3. Material Characterization

The crystal structure and composition were determined by X-ray diffraction (XRD, Ultima IV, using Cu Kα) at a scan rate of 10° min^−1^. The morphology was characterized by scanning electron microscopy (SEM, Hitachi SU8000, Hitachi, Tokyo, Japan). The microstructure was characterized by transmission electron microscopy (TEM, JEOL-2100F Plus, JEOL, Osaka, Japan). Raman spectroscopy was performed on a Raman spectrometer system (JY HR-800 Lab Ram), with a laser wavelength of 532 nm. The specific surface area, pore volume, and pore size distribution of the materials were tested by a specific surface area and porosity analyzer (BET, Micromeritics ASAP 2460, Norcross, GA, USA), with nitrogen adsorption and desorption tests at 77.3 K.

### 3.4. Electrochemical Characterization

The Si@G or EG, binder carboxymethyl cellulose (CMC), and conductive agent Super P were weighed in the ratio of 7:2:1, mixed and ground, and then anhydrous ethanol and deionized water was added in a 1:1 solution as a solvent and stirred for 12 h. After that, the resulting slurry was daubed on the aluminum foil, dried under vacuum at 80 °C for 12 h, and cut into 14 mm diameter round pole pieces after drying.

The CR2032 button cell was made in an argon-filled glove box. Half-cell was assembled to evaluate the lithium storage capacity of Si@G composite. Si@G was used as working electrodes. Lithium metal was used as both counter and reference electrode. Celgard-2400 was used as the separator. In total, 1 M LiPF_6_ and 10 wt% fluoroethylene carbonate (FEC) in a 1:1:1 (volume ratio) mixture of ethylene carbonate (EC), diethyl carbonate (DEC), and dimethyl carbonate (DMC) was used as the electrolyte. In full-DIB assembly, EG was used as the cathode, Si@G was used as anode, and the electrolyte was changed to 4 M LiPF_6_ and 2.0% VC in the ethyl methyl carbonate (EMC). The higher lithium salt concentration induced the solvent molecules to complex with LiPF_6_. At high voltage, the complexed solvent molecules had stronger antioxidant properties, and the electrolyte was more stable. In addition, the VC in the electrolyte as an additive could preferentially form a SEI film with excellent performance on the graphite electrode, which could effectively inhibit the continuous decomposition of solvent molecules on the electrode surface. Cyclic voltammetry (CV) was performed on ModuLab XM ECS-type electrochemical workstation at scan rates of 0.5 mV s^−1^ between 3.00–5.00 V. Charge/discharge tests (GCD) at different current densities at 3.00–5.00 V were performed with the NEWARE 8.0 battery test system. Electrochemical impedance spectroscopy (EIS) was performed on a ModuLab XM ECS-type electrochemical workstation, with a frequency of 0.01 Hz–100 kHz and an amplitude of 5 mV.

## 4. Conclusions

In summary, we prepared a silicon and graphene composite (Si@G) anode through in-situ electrostatic self-assembly and a post-annealing reduction process and investigated it as an anode in full DIBs, together with home-made expanded graphite (EG) as a fast kinetic cathode. The composite structure could suppress the stress/strain induced by volume change and alleviated the pulverization during charging and discharging. In addition, the graphene layer adhered on the surface of Si nanospheres could weaken volume expansion, prevent the inter-agglomeration of Si, and improve the conductivity. Benefiting from that, the as-prepared Si@G anode could retain a maximum specific capacity of 1182.4 mAh g^−1^ after 100 cycles, while the bare Si anode only maintained at 435.8 mAh g^−1^. Moreover, the full Si@G //EG DIBs achieved a high energy density of 367.84 Wh kg^−1^ at a power density of 855.43 W kg^−1^. Thus, this work shed some light on the practical applications of high energy DIBs.

## Figures and Tables

**Figure 1 molecules-28-04280-f001:**
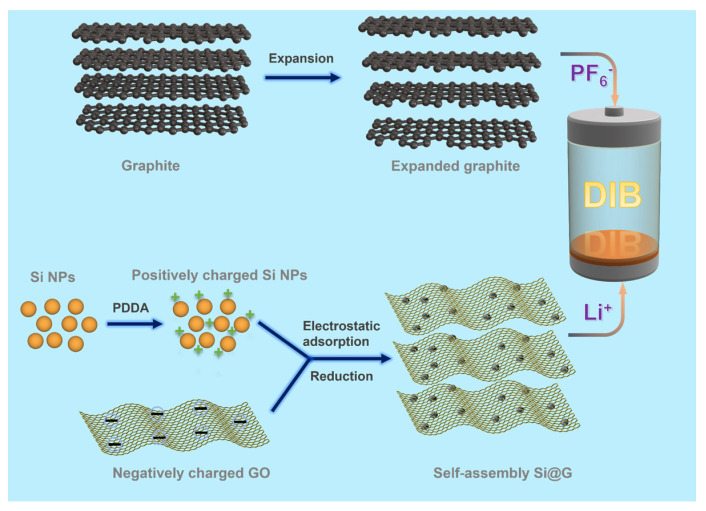
Diagram of the preparation process.

**Figure 2 molecules-28-04280-f002:**
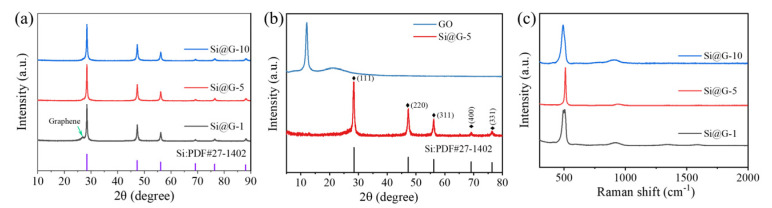
(**a**) XRD comparison of Si@G-1, Si@G-5, and Si@G-10, (**b**) XRD of Si@G-5 and GO, (**c**) Raman plot comparison of Si@G-1, Si@G-5, and Si@G-10.

**Figure 3 molecules-28-04280-f003:**
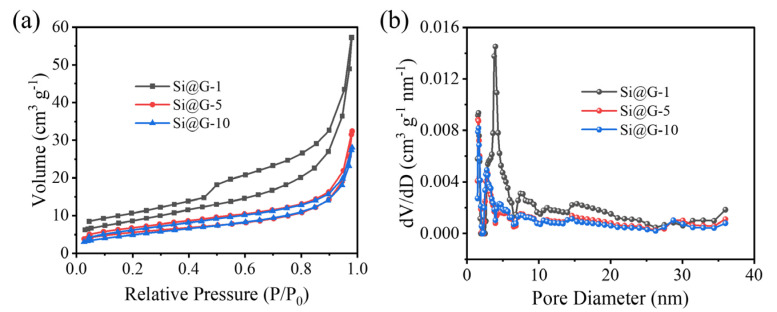
(**a**) Comparison of BET nitrogen adsorption and desorption curves for Si@G-1, Si@G-5, and Si@G-10 samples, (**b**) pore diameter distribution.

**Figure 4 molecules-28-04280-f004:**
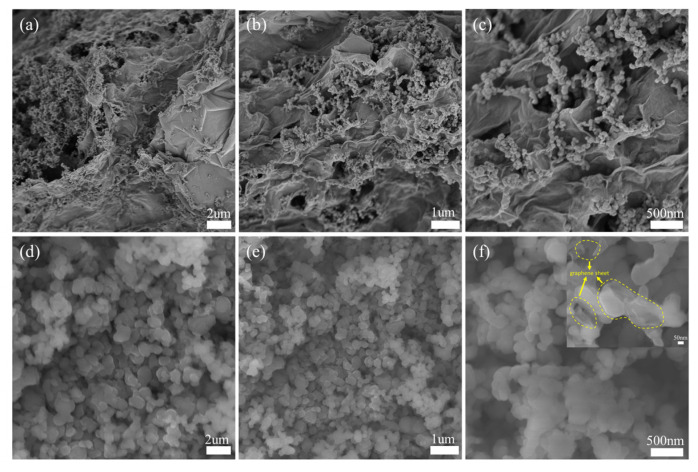
SEM comparison at different magnifications: (**a**–**c**) Si@G-1, (**d**–**f**) Si@G-5.

**Figure 5 molecules-28-04280-f005:**
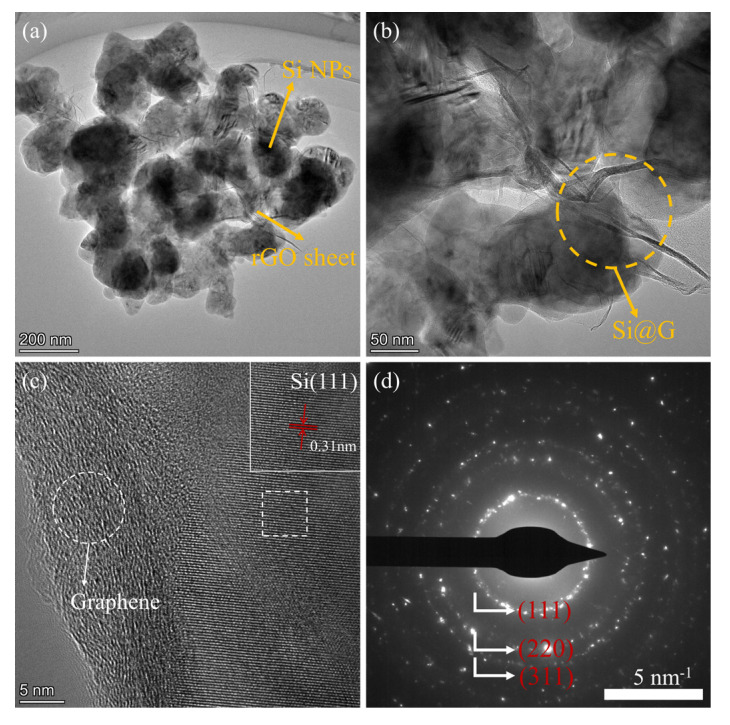
(**a**,**b**) TEM images of Si@G-5 material at different magnifications, (**c**) HRTEM of silicon–graphene composites, (**d**) Dark-field diffraction images of Si@G-5.

**Figure 6 molecules-28-04280-f006:**
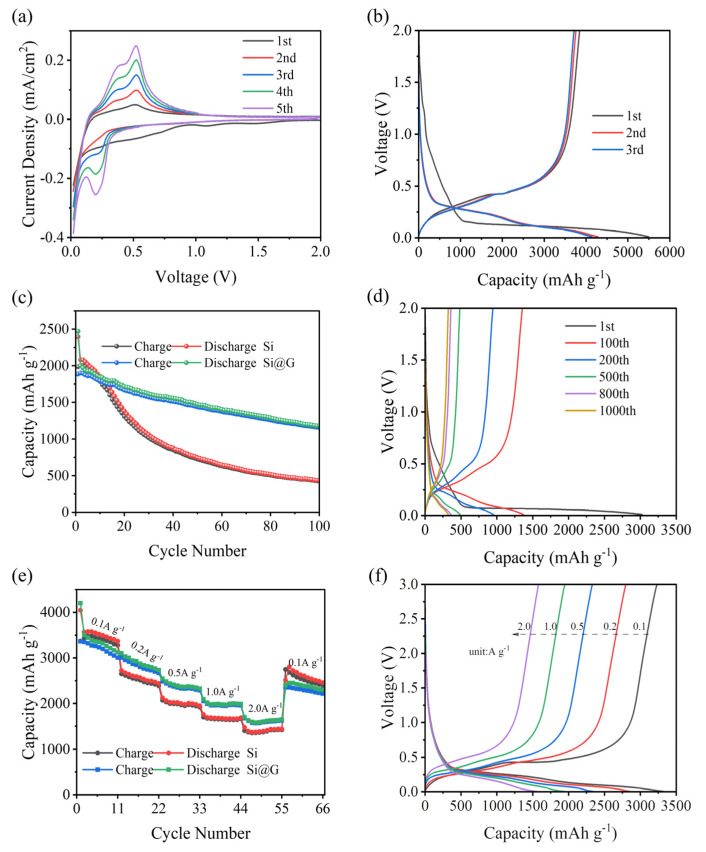
(**a**) CV curves of the first five cycles of the Si@G-5 electrode at 0.5 mV s^−1^, (**b**) Charge-discharge curves of the first three cycles of the Si@G-5 electrode at 0.1 A g^−1^, (**c**) GCD comparisons between pure silicon and Si@G-5 at 0.2 A g^−1^, (**d**) GCD curves of Si@G-5 at 1 A g^−1^ for different cycles, (**e**) Comparison of rate performance between pure silicon and Si@G-5, (**f**) Charge–discharge curves of Si@G-5 at different current densities.

**Figure 7 molecules-28-04280-f007:**
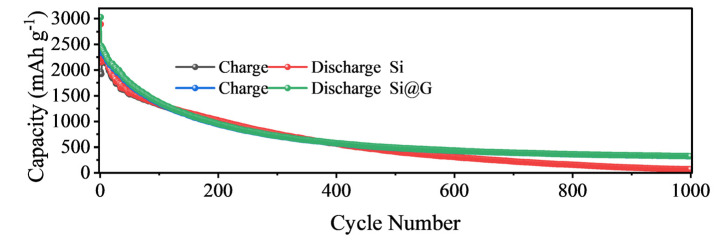
Comparison of long-term cycle performance of pure Si and Si@G-5 at 1.0 A g^−1^.

**Figure 8 molecules-28-04280-f008:**
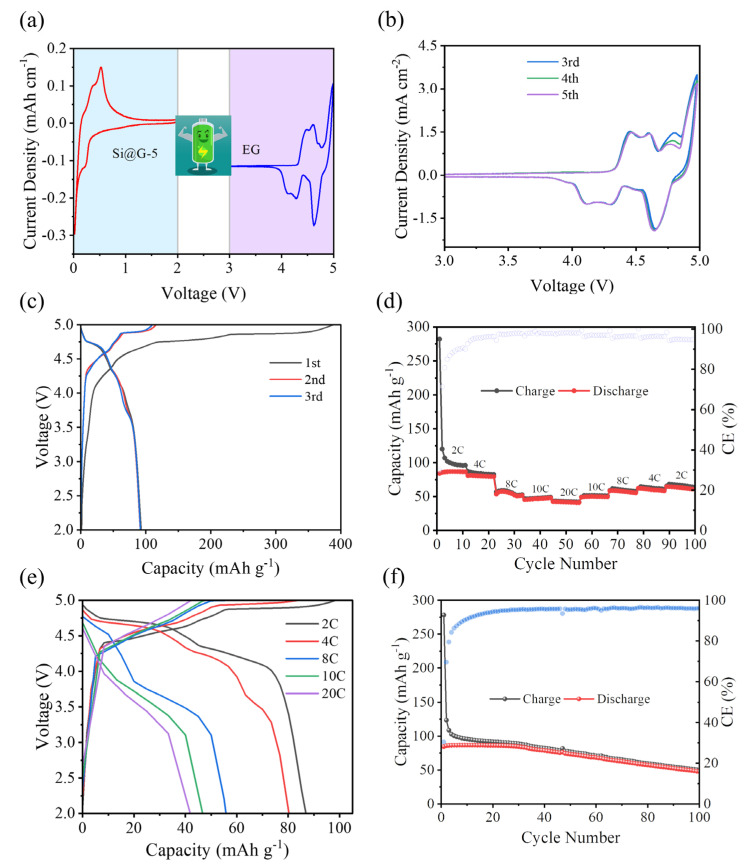
Electrochemical performance of EG//Si@G-5 DIB: (**a**) selection of voltage range, (**b**) CV curves at 0.5 mV s^−1^, (**c**) the first three GCD curves at 1 C, (**d**) rate performance, (**e**) GCD curves at different current densities, (**f**) 100 cycles performance at 2 C.

**Figure 9 molecules-28-04280-f009:**
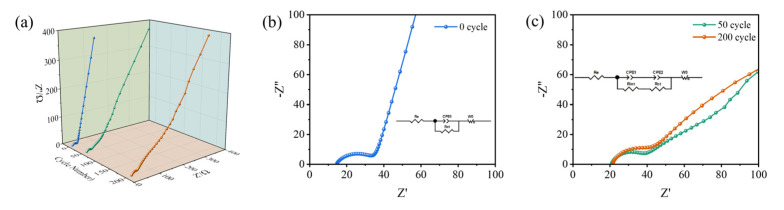
EG//Si@G-5 DIB kinetic analysis: (**a**) EIS comparison at different cycle times, (**b**) EIS without cycle, (**c**) EIS comparison after 50 and 200 cycles (The inset is the equivalent circuit diagram).

## Data Availability

Data will be made available on request.

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
