# Peer review of "High-Performance Dual-Ion Battery Based on Silicon–Graphene Composite Anode and Expanded Graphite Cathode"

_molecules, 2023, doi:10.3390/molecules28114280_

Round 1

Reviewer 1 Report

The manuscript deals with the anode of dual-ion batteries using silicon-graphene composites. The main concern is on the fundamental parameter determining the capacity of this system. Though a higher capacity electrode is implemented, the capacity of dual-ion batteries is defined by the solubility of the salts in the electrolyte. This reviewer does not see how the capacity of this type of battery can be improved by implementing a high capacity anode.

The introduction consists of two paragraphs. The first paragraph is too long. The problem statement is unclear. The research gap is given however it is skeptical if a high capacity anode can really improve the capacity.

Capacity fading is quite high. Also, the authors need more material characterization to support the study. Without appropriate experimental design and set up, the contribution is marginal. We do not learn much from the work.

This reviewer does not have comments on the English.

Author Response

Response to Reviewer 1 Comments

Reviewer 1#

The manuscript deals with the anode of dual-ion batteries using silicon-graphene composites. The main concern is on the fundamental parameter determining the capacity of this system. Though a higher capacity electrode is implemented, the capacity of dual-ion batteries is defined by the solubility of the salts in the electrolyte. This reviewer does not see how the capacity of this type of battery can be improved by implementing a high capacity anode.

The introduction consists of two paragraphs. The first paragraph is too long. The problem statement is unclear. The research gap is given however it is skeptical if a high capacity anode can really improve the capacity.

Capacity fading is quite high. Also, the authors need more material characterization to support the study. Without appropriate experimental design and set up, the contribution is marginal. We do not learn much from the work.

Response: Thank you very much for your constructive comments, which are beneficial to the improvement of the manuscript.

We agree that the capacity of DIBs is highly dependent on the salt concentration in the electrolyte. However, the choice of anode is also an important issue in DIBs because it affects the performance of the battery (i.e., rate capacity, cycling stability, specific capacity, safety, etc.) together with the cathode material and other components in the battery. It was found that the replacement of low capacity of graphite anode with other large capacity anode could indeed increase the capacity of the full DIBs. For instance, Wei et al. [ ACS Sustainable Chemistry & Engineering 2020, 8, (14), 5514-5523] assembled DIBs employing high lithium storage capacity of MoSe2/Nitrogen-Doped Carbon (1224 mA h g-1) as anode and graphite as cathode, and the DIBs delivered a reversible discharge capacity of 86 mA h g–1 at 2C after 150 cycles. Tang et al. [ Advanced Energy Materials 2016, 6, (11), 1502588] studied Al foil (theoretical lithium storage capacity is 2235 mAh g−1) as anode, when coupled with graphite cathode, the DIBs showed a reversible capacity of ≈100 mAh g−1 and a capacity retention of 88% after 200 charge–discharge cycles. Wang et al. [ Nanoscale 2020, 12, (1), 79-84] designed a high capacity of Ge/CNFs anode (2614 mA h g−1), and the assembled Ge/CNFs-Graphite DIBs showed a high discharge capacity of 281 mA h g−1 at a discharge current of 0.25 A g−1, which greatly surpasses those of most of the reported DIBs. Therefore, in order to take full advantage of DIBs, further optimization of anode materials is also an important way.

Therefore, we applied the idea of alloying materials with high lithium storage capacity to DIBs. Silicon anode, has an ultra-high theoretical specific capacity (4200 mAh g-1), suitable potential and abundant storage capacity. However, the huge volume expansion and poor electrical conductivity of Si hinders its practical application. Therefore, we prepared a strongly coupled silicon and graphene composite (Si@G) anode and the merits of the composite electrode lies that: (â…°) the composite structure greatly suppressed the stress/strain induced by volume change and alleviated the pulverization during charging and discharging; (â…±) the graphene layer adhered on the surface of Si nanospheres could retard volume expansion and prevent the inter-agglomeration of Si with high surface energy; (â…²) the electron migration during the charge/discharge process was promoted by highly conductive graphene; (â…³) the improved rate ability of Si@G anode matched well with the fast kinetics of EG cathode.

The capacity fading of DIBs is usually high which can be attributed to the following: firstly, the unstable SEI consumes a substantial amount of active components during the formation process. Secondly, DIBs also suffer from certain oxidative decomposition and co-intercalation of electrolyte salts at high voltages leading to irreversible capacity increases. Finally, exfoliation of the graphite cathode sheets during repeated intercalation also deteriorates the performance. We will continue to do the research of improving the cycling performance of DIBs.

We have split the first paragraph of introduction into two paragraphs and illustrated the opinion that high capacity anode can make good effect on improving the battery capacity with several reported research papers.

Changes:In line 31-76,

“Despite the high energy and power density of lithium ion batteries, the limited and uneven distribution of lithium and rare metal resources have led to the search of new energy storage technologies with low cost, high safety and reliability [1-3]. DIBs are energy storage devices which store energy involving the intercalation of both anions and cations on the cathode and anode simultaneously, and featured high output voltage, low cost and good safety [4]. Graphite was commonly used as cathode electrode because it could accommodate intercalation of anions (i.e., PF6-, BF4-, ClO4-) at high cut-off voltage (up to 5.2 V vs Li+/Li) [5, 6]. However, the tested capacity of graphite cathode is usually ranged between 80-130 mAh g-1, and it is difficult to be further increased. As for the anode side, any materials which can reversibly store cations can be used as anodes for DIBs [7]. In this case, the traditional graphite anode could be exchanged to other electrode materials with larger capacity. Recently, it was found that the replacement of low capacity of graphite anode with other large capacity anode could indeed increase the capacity of the full DIBs. For instance, Wei et al. [8] assembled DIBs employing high lithium storage capacity of MoSe2/Nitrogen-Doped Carbon (1224 mA h g-1) as anode and graphite as cathode, and the DIBs delivered a reversible discharge capacity of 86 mA h g–1 at 2C after 150 cycles. Tang et al. [9] studied Al foil (theoretical lithium storage capacity is 2235 mAh g−1) as anode, when coupled with graphite cathode, the DIBs showed a reversible capacity of ≈100 mAh g−1 and a capacity retention of 88% after 200 charge–discharge cycles. Wang et al. [10] designed a high capacity of Ge/CNFs anode (2614 mA h g−1), and the assembled Ge/CNFs-Graphite DIBs showed a high discharge capacity of 281 mA h g−1 at a discharge current of 0.25 A g−1, which greatly surpasses those of most of the reported DIBs. Therefore, in order to take full advantage of DIBs, further optimization of anode materials is also an important way.

According to the cation storage mechanisms, the anode materials can be divided to three types, as intercalation-type, conversion-type and alloying-type. However, the intercalation-type anode (such as graphite, soft carbon, hard carbon, Na2Ti3O7, etc.) usually exhibit low capacity and the conversion-type anode (such as MoS2, MoSe2, Co3O4, etc.) suffer from low reaction kinetics and show unimpressed rate performance. The alloying-type anode (such as Al, P, Si, Sn, Ge, etc.) which can react with cations and boost an extreme specific capacity [11-13]. Silicon anode materials possess a high theoretic Lithium storage capacity of 4200 mAh g-1 [14-16]. Therefore, it is an efficient method to improve the energy density of DIBs by combining graphite cathode with high-capacity silicon anode. However, the huge volume expansion (>300%) of silicon throughout alloying and de-alloying must be controlled [17-20]. Moreover, the poor electrical conductivity of silicon is also an important factor that hinders its development [21-23]. Up to now, there are only a few reports about exploring Si as anode in DIBs. Shao et al. [24] prepared a Si/C anode by adding a little amount of Si (7.6 wt%) into graphite and used it a full DIBs with enhanced energy density. Wang et al. [25] investigated a pre-lithiated Si as anode with the regards of tailoring the voltage to match the cathode. Tang et al. [26] cleverly designed a flexible interface between Si and a conducting soft polymer substrate to modulate the alloying stress of the silicon anode in DIBs and achieved excellent flexible electrochemical properties. Although the above efforts, to fully take advantage of the large capacity of Si anode, it is still a challenge to further design new Si based anode material with controlled volume expansion effect and improved conductivity as well as matched kinetics with the cathode side.”

Reviewer 2 Report

Dear Authors,

the manuscript present the work on Si incorporation in the negative electrode of (full) dual ion batteries. Results from half cells and from full cells are presented. The working electrodes of half cells, and the negative and positive electrodes of the full cells possed some 30% of binder and conductive agents. The working electrode of half cells are silicon, expanded graphite, and Si-nanoparticles embedded in graphene sheets. Expanded graphite was used also for the positive electrode (cathode) of the dual ion battery (DIB). The positive electrode of the full lithium-ion battery (LIB) contained LiFePO4. The negative electrode of the DIB and LIB possess Si-nanoparticles embedded in graphene sheets. The work is of interest for the battery community and merits therefore publication. Before doing so, there are the following points (1-to-12) to be rectified. 

1) Of what material is the counter electrode of the half cells? Lithium metal?

2) The anions intercalation potentials are in between 4 V and 5 V in respect to the Li-metal reference electrode (e.g., Figure 8). The electrolyte is unstable at such a high potential. Please discuss this in your manuscript. 

3) There are no results of the half cell which working electrode is expanded carbon for anion intercalation. Why?.

4) Figures 1,2,3,6,8 and 9 are to small. Please plot them larger, i.e., by rearranging the diagrams.

5) at line 117 (XRD): there should be 100 min-1 instead of 100C min-1.

6) lines 129-134: Please split the long sentence in two or three shorter sentences.

7) Please describe (e.g., through equations in the supplementary materials) how you determined the mass, the specific capacity, the full cell capacity, the energy density, and the power density. 

8) line 179: concerning the following sentence: ... silicon nanoparticles, whose size ranges from 1 to 10 um. What is um? From SEM (Figure 4f) I would say that the nanoparticles size is aproximatively 100 nm.

9) lines 186-187: I cannot see the graphene sheets in SEM.

10) lines 240-242: I cannot anderstand the sentence on that lines. What do you mean with: plattform remains inhibited? Use please other words.

11) line 281: ... EIS was (and not were) performed.

12) line 281-288: Why should flatter low-frequency parts of the impedance  indicate lower Li diffusivity? Please discuss in more detail and give more exact values about Li diffusivity from the analysis of the presented EIS data.

Best regards,

the reviewer

The English is OK. Some improvements can be done.

Round 2

Reviewer 1 Report

The revised version is acceptable for publication.